# Aseptic Abscess Syndrome: Clinical Characteristics, Associated Diseases, and up to 30 Years’ Evolution Data on a 71-Patient Series

**DOI:** 10.3390/jcm11133669

**Published:** 2022-06-25

**Authors:** Ludovic Trefond, Camille Frances, Nathalie Costedoat-Chalumeau, Jean-Charles Piette, Julien Haroche, Laurent Sailler, Souad Assaad, Jean-François Viallard, Patrick Jego, Arnaud Hot, Jerome Connault, Jean-Marc Galempoix, Elisabeth Aslangul, Nicolas Limal, Fabrice Bonnet, Stanislas Faguer, Olivier Chosidow, Christophe Deligny, François Lifermann, Alexandre Thibault Jacques Maria, Bruno Pereira, Olivier Aumaitre, Marc André

**Affiliations:** 1Médecine Interne, CHU Gabriel Montpied, 63000 Clermont-Ferrand, France; oaumaitre@chu-clermontferrand.fr (O.A.); mandre@chu-clermontferrand.fr (M.A.); 2M2iSH, UMR 1071 Inserm, INRA USC 2018, University of Clermont Auvergne, 63000 Clermont-Ferrand, France; 3Faculté de Médecine, AP-HP, Service de Dermatologie et Allergologie, Hôpital Tenon, Sorbonne Université, 75252 Paris, France; camille.frances@me.com; 4APHP, Service de Médecine Interne, Centre de Référence des Maladies Auto-Immunes Systémiques Rares d’Ile de France, Hôpital Cochin, 27 rue du Faubourg St-Jacques, CEDEX 14, 75679 Paris, France; nathalie.costedoat@aphp.fr; 5INSERM U 1153, Centre of Research in Epidemiology and Statistics Sorbonne Paris Cité (CRESS), Université de Paris, 75006 Paris, France; 6Service de Médecine Interne, AP-HP Groupe Hospitalier Pitié-Salpêtrière, 75013 Paris, France; jcpiette@free.fr; 7Assistance Publique–Hôpitaux de Paris (AP-HP), Groupement Hospitalier Pitié–Salpêtrière (GHPS), French National Reference Center for Systemic Lupus Erythematosus, Antiphospholipid Antibody Syndrome and Other Autoimmune Disorders, Service de Médecine Interne 2, Institut E3M, Sorbonne Université, 75252 Paris, France; julien.haroche@aphp.fr; 8Internal Medicine Department, CHU de Toulouse—Hôpital Purpan, 31300 Toulouse, France; sailler.l@chu-toulouse.fr; 9Centre Léon Bérard, 69008 Lyon, France; souad.assaad@lyon.unicancer.fr; 10Hôpital Haut-Lévêque, CHU de Bordeaux, Service de Médecine Interne et Maladies Infectieuses, Université de BORDEAUX, 5 Avenue de Magellan, 33604 Pessac, France; jean-francois.viallard@chu-bordeaux.fr; 11Irset (Institut de Recherche en Santé, Environnement et Travail)-UMR_S 1085, Inserm, EHESP, University of Rennes, 35000 Rennes, France; patrick.jego@chu-rennes.fr; 12Department of Internal Medicine, Rennes University Hospital, 35203 Rennes, France; 13Service de Médecine Interne, Hôpital Edouard Herriot, Hospices Civils de Lyon, 69437 Lyon, France; arnaud.hot@chu-lyon.fr; 14Department of Internal and Vascular Medicine, CHU de Nantes, 44000 Nantes, France; jerome.connault@chu-nantes.fr; 15Médecine Infectieuse, CHInA, 08000 Charleville-Mézières, France; jean-marc.galempoix@ch-nord-ardennes.fr; 16Service de Médecine Interne, Hôpital Louis-Mourier, Assistance Publique-Hôpitaux de Paris, 92701 Colombes, France; elisabeth.aslangul@aphp.fr; 17UPD5, Université Paris-Descartes, rue de l’École-de-Médecine, 75006 Paris, France; 18Département de Médecine Interne, Hôpital Henri Mondor, APHP Université Paris-Est Créteil, 94010 Créteil, France; nicolas.limal@aphp.fr; 19Department of Internal Medicine and Infectious Diseases, Bordeaux University Hospital, Saint André Hospital, 33000 Bordeaux, France; fabrice.bonnet@chu-bordeaux.fr; 20Département de Néphrologie et Transplantation d’Organes, Centre de Référence des Maladies Rénales Rares, CHU de Toulouse, 31000 Toulouse, France; faguer.s@chu-toulouse.fr; 21Department of Dermatology, APHP, Hôpital Henri-Mondor, 94010 Créteil, France; olivier.chosidow@aphp.fr; 22Research Group Dynamic, EA7380, Faculté de Santé de Créteil, Ecole Nationale Vétérinaire d’Alfort, USC ANSES, Université Paris-Est Créteil, 94010 Créteil, France; 23Service de Médecine Interne, CHU de Fort de France, 97200 Fort de France, France; christophe.deligny@chu-martinique.fr; 24Medecine Interne, Centre Hospitalier de Dax, 40107 Dax, France; lifermannf@ch-dax.fr; 25Department of Internal Medicine, Saint Eloi Hospital, Montpellier University, 34000 Montpellier, France; a-maria@chu-montpellier.fr; 26Biostatistics Unit (DRCI), University Hospital Clermont-Ferrand, 63000 Clermont-Ferrand, France; bpereira@chu-clermontferrand.fr

**Keywords:** aseptic abscess syndrome, Crohn’s disease, colchicine, biologics

## Abstract

Aseptic abscess (AA) syndrome is a rare type of inflammatory disorder involving polymorphonuclear neutrophils (PMNs), often associated with inflammatory bowel disease (IBD). This study sought to describe the clinical characteristics and evolution of this syndrome in a large cohort. We included all patients included in the French AA syndrome register from 1999 to 2020. All patients fulfilled the criteria outlined by André et al. in 2007. Seventy-one patients were included, 37 of which were men (52.1%), of a mean age of 34.5 ± 17 years. The abscesses were located in the spleen (71.8%), lymph nodes (50.7%), skin (29.5%), liver (28.1%), lung (22.5), and rarer locations (brain, genitals, kidneys, ENT, muscles, or breasts). Of all the patients, 59% presented with an associated disease, primarily IBD (42%). They were treated with colchicine (28.1%), corticosteroids (85.9%), immunosuppressants (61.9%), and biologics (32.3%). A relapse was observed in 62% of cases, mostly in the same organ. Upon multivariate analysis, factors associated with the risk of relapse were: prescription of colchicine (HR 0.52; 95% CI [0.28–0.97]; *p* = 0.042), associated IBD (HR 0.57; 95% CI [0.32–0.99]; *p* = 0.047), and hepatic or skin abscesses at diagnosis (HR 2.14; 95% CI [1.35–3.40]; *p* = 0.001 and HR 1.78; 95% CI [1.07–2.93]; *p* = 0.024, respectively). No deaths occurred related to this disease. This large retrospective cohort study with long follow up showed that AA syndrome is a relapsing systemic disease that can evolve on its own or be the precursor of an underlying disease, such as IBD. Of all the available treatments, colchicine appeared to be protective against relapse.

## 1. Introduction

Aseptic abscess (AA) syndrome is a rare inflammatory disorder involving polymorphonuclear neutrophils (PMNs) of unknown aetiology, which are characterised by circumscribed, PMN-rich, sterile lesions. The first cases were described in the 1990s [1,2]. In 2007, 30 cases were described, 70% of which were associated with inflammatory bowel disease (IBD), with relapses in 60% [3]. Since then, isolated cases have been reported describing rare locations and the effect of new treatments [4,5,6,7,8,9] including biologics, notably anti-interleukin 1 (IL-1) [10,11]. The use of positron emission tomography (PET) has also changed the management of the disease [12].

Despite the availability of diagnostic criteria [3], no case series has been published since 2007. Since the development of several new treatments and diagnostic tools, we no longer have a clear picture of how the syndrome evolves in terms of recurrence, associated disease, and management.

This study sought to describe the clinical, paraclinical, therapeutic characteristics, and risks factors for the relapse of AA syndrome, along with its evolution, based on a multicentric French cohort from 1999 to 2020.

## 2. Materials and Methods

All patients registered in the French AA syndrome register (Freedom of Information Act—Commission nationale de l’informatique et de la liberté—99.149, June 1999) from 1999 to 2020 were included. New French cases were added to the register once reported by physicians and/or through the members of the French National Society for Internal Medicine (Société Nationale Française de Médecine Interne—SNFMI). The study was approved by the local Ethics Committee (IRB00013412, “CHU de Clermont Ferrand IRB #1”, IRB number 2022-CF020) with compliance to the French policy of individual data protection. The patients’ evolution was recorded from medical records and by communication with the physicians or patients themselves via email or telephone. A total of 20 patients belonged to the 2007 case series [3] and their updated evolution data were collected. All patients consented to be included in the register and have their data collected. The inclusion criteria were those proposed by André et al. in 2007 [3]: (1)Deep abscess(es) on imaging, with a histological predominance of PMNs if puncture or biopsy was performed;(2)Negative blood cultures; negative serological tests, notably for Yersinia enterocolitica; and if puncture or biopsy was performed: standard bacteriology, acid-alcohol fast bacilli (AAFB) test, mycology, and pus parasitology all negative;(3)Antibiotic failure, if prescribed, following at least 2 weeks’ treatment for conventional antibiotherapy and 3 months for anti-tuberculosis drugs;(4)Fast clinical improvement observed the day after administering corticosteroids (CS), followed by radiological improvement after 1 month of CS, sometimes associated with immunosuppressants.

We collected epidemiological, clinical, biological, radiological, histological, and evolution data. Diagnosis of associated diseases was left to the discretion of each patient’s primary physician. Relapse was defined as the occurrence of new abscesses, either clinical or revealed on imaging, with or without other symptoms (such as fever or pain) and/or biological signs (increased CRP or PMN hyperleukocytosis) following a period of clinical remission.

Continuous data were expressed according to statistical distribution as mean and standard deviation or median and interquartile range. The assumption of normality was assessed by the Shapiro–Wilk test. Categorical parameters were compared between groups using chi-squared or Fisher’s exact tests, whereas continuous variables were compared between groups by Student’s *t*-test or the Mann–Whitney test when the assumptions of the *t*-test were not met. The homoscedasticity was analysed using the Fisher–Snedecor test. 

Estimates of relapse-free survival were constructed using the Kaplan–Meier method. As a patient could present several relapses, the marginal Cox proportional hazards regression model for repeated data was used to investigate associated prognostic factors in univariate and multivariable analysis, taking into account between and within patient variability. For multivariable analysis, the covariates were determined according to univariate results and to the clinical relevance with a particular attention paid on multicollinearity, (1) studying the relationships between the covariables and (2) evaluating the impact to add or delete variables on the multivariable model. IBD, relapsing polychondritis (RP), and pyoderma gangrenosum were treated independently in multivariable analyses. The proportional hazard hypothesis was verified using Schoenfeld’s test and plotting residuals.

To ensure the robustness of our results, the final model was validated by a two-step bootstrapping process. In each step, 1000 bootstrap samples with replacements were created from the training set. In the first one, using the stepwise procedure, we determined the percentage of models including each of the initial variables. In the second step, we independently estimated the marginal Cox model parameters of the final model. The bootstrap estimates of each covariate coefficient and standard errors were averaged from those replicates. Results were expressed as a hazard ratio (HR) and 95% confidence interval. 

Statistical analysis was performed using Stata software (version 15, StataCorp LP, College Station, TX, USA). The tests were two-sided, with a type I error set at 5%. 

## 3. Results

This section is divided by subheadings. It provides a concise and precise description of the experimental results, their interpretation, and the experimental conclusions that can be drawn.

### 3.1. Characteristics of the Patients

We included 71 patients, 37 of which were men (52.1%). Mean age at diagnosis was 34.5 ± 17 years (Table 1).

#### 3.1.1. Manifestations on Diagnosis

None of the patients had arterial hypotension, though 88.2% presented with fever. Mean temperature of febrile patients was 39.4 ± 0.65 °C. Forty-six patients (64.7%) had abdominal pain without guarding or rebound tenderness. Skin manifestations were present in 49% of the patients, including: abscesses (25.3%), oral aphthosis (16.9%), erythema nodosum (7.0%), pustules (8.5%), and Sweet syndrome (2.8%). Palpable splenomegaly was present in nine patients (12.6%). The median PMN count was 15,100 per mm^3^ (12,000–19,750). The median CRP was 120 mg/L (67–234). 

Colonic macroscopic or histological abnormalities were reported in 9 out of 30 patients who had undergone a colonoscopy at AA syndrome diagnosis (3 patients without digestive symptoms).

#### 3.1.2. Manifestations during the Course of the Disease

Of the 18 patients presenting with abscesses limited to the spleen on diagnosis, 14 had no extra-splenic involvement during the disease course. Lymph node AAs were found in 36 patients (50.7%), in the following locations: intra-abdominal (*n* = 31), cervical (*n* = 2), mediastinal (*n* = 2), and inguinal (*n* = 1). Four patients had brain abscesses occurring only at relapse, which were either intra-parenchymatous (*n* = 2) or pituitary (*n* = 2). Those affecting the genitals were in the vagina (*n* = 3), testicles (*n* = 1), or prostate (*n* = 1). Seven patients presented with muscular abscesses located as follows: quadriceps (*n* = 3), psoas (*n* = 2), and cervical muscles (*n* = 2). The mean number of organs affected was 1.3 ± 1.6 at diagnosis versus 2.5 ± 1.5 during the entire disease course. In addition to cutaneous neutrophilic manifestations, other systemic aseptic neutrophilic manifestations were found during the course of the disease: meningitis (3), neutrophilic pleural effusion (2), myocarditis (*n* = 1), and ascites (*n* = 1). Nineteen patients (26.7%) had a PET scan at diagnosis or relapse and all had at least one hypermetabolic AA syndrome location.

Histopathology confirmation was obtained in 59 patients (83.1%). Abscess puncture was performed in 40 patients (56.3%) in the following locations: spleen (*n* = 25), abdominal lymph nodes (*n* = 15), skin (*n* = 3), liver (*n* = 2), superficial lymph nodes (*n* = 1), tongue (n = 1), muscle (*n* = 1), lung (*n* = 1), kidney (*n* = 1), and breast (*n* = 1). Splenectomy was performed in 23 patients. Gigantocellular epithelioid granulomas without caseous necrosis was described in 19 patients (32.2%).

A splenectomy was performed in 23 patients (32.3%), on average 7.2 ± 15.2 months after diagnosis. The procedure led to an improvement of the symptoms in half of the patients but all relapsed later. On relapse, abscesses were located in lymph nodes (*n* = 7), liver (*n* = 6), lung (*n* = 4), and brain (*n* = 4) (Appendix A).

### 3.2. Associated Conditions

An inflammatory disease was associated with the AA syndrome in 41 patients (59.4%), diagnosed before (*n* = 18), concomitantly (*n* = 7), or after (*n* = 16) the AA syndrome (Table 2). IBD was associated with AA syndrome in 30 patients (42.2) and was diagnosed on average 0.7 ± 5.2 (min −10, max + 16) years after AA syndrome. We compared the characteristics of patients with and without IBD (Appendix A). The only significant difference was the age: 26.9 ± 1.77 in the IBD group and 40.1 ± 3.02 in the non-IBD group (*p* < 0.001).

### 3.3. Treatments

Antibiotics were prescribed to 59 patients (83.1%) and 11 received anti-tuberculosis treatment, with no success. CSs were prescribed to 57 patients (80.2%) at diagnosis (mean dose: 0.7 ± 0.44 mg/kg). Time to fever resolution following CS treatment was ≤ 24 h (data available in 15 patients). Six patients (8.4%) received only colchicine upon diagnosis, two of whom did not relapse. An immunosuppressant drug was added upon diagnosis in 16 patients (22.5%) and in 44 patients (61.9%) during the course of the disease. The drugs prescribed upon diagnosis and upon relapse are reported in Appendix A. The main immunosuppressant drugs were: azathioprine (*n* = 22; 30.9%), cyclophosphamide (*n* = 8; 11.2%), and methotrexate (*n* = 3; 4.2%), and regarding the biologics, infliximab (*n* = 14; 19.7%), adalimumab (*n* = 8; 11.2%), and anakinra (*n* = 4; 5.6%).

### 3.4. Evolution

Abscesses relapsed at least once in 61% of patients (*n* = 44) (Table 3). The median relapse-free survival was 0.8 years (0.50–2.91). On relapse, 25 patients were on treatment: CSs (*n* = 25), immunosuppressants (*n* = 9), and biologics (*n* = 6). The median dose of CSs at the time of relapse was 12.5 (10;20) milligrams. Six patients became pregnant after the AA diagnosis, none of whom experienced a disease flare during pregnancy.

One patient relapsed one month after initiation of nivolumab for pulmonary cancer. Aseptic abscesses syndrome was so far controlled under a low dose of CSs and relapsed on the spleen, while the pulmonary cancer was still in remission. We looked for risk factors for relapse regarding general characteristics, location of abscesses, associated disease, or treatment (Table 4). Hepatic abscess at diagnosis (HR 2.14; 95% CI (1.35–3.40); *p* = 0.001) and skin abscesses at diagnosis (HR 1.78; 95% CI (1.07–2.93); *p* = 0.024) were associated with an increased risk of relapse. Colchicine (HR 0.52; 95% CI (0.28–0.97); *p* = 0.042) and associated IBD (HR 0.57; 95% CI (0.32–0.99); *p* = 0.047) were associated with a lower risk of relapse. 

There were five deaths recorded, due to multiple causes: ischemic cardiomyopathy, viral hepatitis, lung cancer, stroke, and haemorrhagic shock following tracheotomy. None of the patients died directly because of aseptic abscesses syndrome but the impact of the CSs and immunosuppressive drug was discussed for ischemic cardiomyopathy and viral hepatitis.

## 4. Discussion

We conducted a nation-based retrospective study presenting the largest cohort of AA syndrome patients and their evolution up to 30 years post-diagnosis. Overall, 41% presented with isolated AA syndrome, while 42% had an associated IBD, which was diagnosed later in half of cases. Disease relapse was reported in 62% of the patients, most commonly in the same organ as before and within the 25 years following diagnosis. Colchicine treatment was a protective factor against relapse.

Given that 41% of patients presented with an isolated AA syndrome, the question of the accuracy of a true AA disease is raised. In the cases with associated diseases, the AAs primarily evolved independently, with no significant difference found between the group with associated diseases and that without. Prior to their AA syndrome diagnosis, only 25% had an associated disease, compared to 59% after 25 years of evolution. AA syndrome thus appears to be more a of a precursor of a future underlying disease than a disease in itself.

Genitourinary abscesses [13] are possible with IBDs, often by contiguity, through fistulas. The five patients of this series presenting with genitourinary involvement did not have an associated IBD. Septic muscle abscesses can develop in Crohn’s disease [14]. In this cohort, we observed aseptic muscle abscesses in five patients, three of whom had a pre-diagnosed IBD, evolving favourably on CT or immunosuppressants.

The patients with solely skin AAs were excluded due to the diagnostic criteria. As in the literature [15,16,17], these abscesses presented differently from those in deep locations, generally without fever, yet the evolution was similar with a positive response to CSs and frequent relapses. As the histopathology description was the same, we can discuss to include this isolated location in the classification criteria.

PET scan was frequently used, with 30% undergoing a scan that revealed at least one hypermetabolic AA location at diagnosis or at relapse. Though PET is not specific, it may be a highly useful tool for differentiating sequelae from active lesions, or for a deeper work-up to search for other abscesses [12]. 

Approximately 8% of IBDs are associated with extra-intestinal manifestations [18]. In our study, those presented with an associated IBD were diagnosed up to 10 years after the diagnosis of AA syndrome. This supports the use of colonoscopy as one of the diagnostic tests [19], which revealed, in our study, morphological or histological abnormalities in 30% of cases when it had been performed, while 10% did not have digestive symptoms. In our study, only mean age at diagnosis of AA syndrome statistically differed between the patients presenting with IBD (26.9 ± 1.77) and those without (40.1 ± 3.02). The biological inflammatory syndromes reported were generally associated with a mean CRP of 149 mg/L, which is higher than in isolated Crohn’s disease, where it is up to 50 mg/L on average at diagnosis [20]. It may be useful in differentiating a Crohn’s disease flare or AA syndrome. Ten patients presented with associated PG (14%). Spleen AAs associated with PG have notably been described in the literature as deep-set PG, despite the descriptions generally fulfilling the criteria of AA syndrome [21,22,23,24,25]. 

Histopathology was performed in most (83%) patients in this cohort. Other patients either had a typical presentation and improved on CS treatment, or a location that was difficult to access by biopsy. The evolution of patients who did not benefit from a histological analysis of their abscesses was not different from the others, but a differential diagnosis should not be overlooked. For example, one patient (not included in our series) with a typical isolated splenic aseptic abscess disclosing 99% of PMN cells at spleen puncture, had a clinical improvement after a course of corticosteroids but liver biopsy disclosed a diagnosis of lymphoma several months later.

Remission was possible without treatment following splenectomy when AAs were limited to the spleen. Authors reported no recurrence for 20 years following splenectomy, although the first lesion was a ruptured splenic abscess, which never occurred in our series [4]. However, in our series, the disease systematically relapsed sooner or later after splenectomy. Therefore, we do not recommend splenectomy early in the disease course once the diagnosis of AA syndrome has been established. 

In 2010, a study on gene transcription in the AA syndromes of seven patients revealed mRNA overexpression of IL-1 beta [26]. In our series, one-third of patients received at least one biologic, primarily anti-TNF and also anti-IL-1 in some of them [10]. Many patients received various biologics but rotating courses of these drugs was necessary and appeared effective in case of relapse. Rarely, it seems also possible to treat without CS treatment, since two patients of our series were treated solely with colchicine and suffered no relapse afterwards. 

Despite the use of new drugs, 61.9% of our patients relapsed, generally less than 1 year after diagnosis. Seventy-two percent of the patients relapsed in the same organ, up to 25 years after diagnosis. In our patients, the classic evolution appeared to be as follows: splenic involvement, potentially including the lymph nodes, then spreading to the liver and finally to other organs (brain, lungs, kidneys, pancreas, genitals, muscles, skin, and breasts), with at least two different affected organs during the disease course in 80% of our patients. AA syndrome thus appears to be a systemic disease. On the other hand, some AA syndrome cases in our series remained limited to the spleen, with 77% of patients with isolated splenic involvement at diagnosis showing no extra-splenic abscesses during the disease course.

We identified factors associated with an increased or decreased risk of relapse, especially the seemingly protective effect of colchicine (HR 0.53 95% CI [0.28–0.99] *p* = 0.048). Colchicine is one of the treatments of choice for auto-inflammatory diseases, known for its effect on PMNs and particularly how they migrate [27,28]. Of note, none of the pregnant patients suffered relapse during pregnancy, despite higher rates of granulocyte colony-stimulating factor (G-CSF) during pregnancy [29]. Still, G-CSF does cause neutrophilic activation; hence, one series reported a patient having received this treatment before developing their disease, along with a case of PG [30]. One patient relapsed while receiving nivolumab. To the best of our knowledge, it was the first case of a patient with AA syndrome receiving an immune checkpoint inhibitor (anti-PD-1/PD-L1). The possibility of inflammatory disease relapse, such as IBDs or Sweet syndrome, after such treatments has been previously described [31,32].

## 5. Conclusions

In conclusion, we present the characteristics and long-term evolution of a large series of patients with aseptic abscess syndrome. This information is important for practitioners and for patients with this rare syndrome. Splenectomy should be avoided when the diagnosis is made and colchicine seems to be an efficacious background treatment. Treatment remains difficult despite the use of currently available biologics.

## Figures and Tables

**Table 1 jcm-11-03669-t001:** Characteristics of 71 patients presenting with aseptic abscess syndrome.

	Total (*n* = 71)
Age (years), mean ± SD	34.5 ± 17
Male gender, *n* (%)	37 (52.1)
Time (months) between 1st symptoms and diagnosis, med [IQR]	5.5 (2.5;20)
Symptoms at diagnosis symptoms, *n* (%)	
Fever	63 (88.2)
Abdominal pain	46 (64.7)
Weight loss	26 (36.6)
Skin manifestations	35 (49.2)
Laboratory abnormalities	
Anaemia	29 (40.8)
PMN leukocytosis	61 (85.9)
Increased CRP	71 (100)
Liver function test abnormalities	18/63 (28.5)
Number of organs involved during the course of the disease, med (min-max)	2 (1–9)
Location of abscesses, *n* (%)	
Spleen	51 (71.8)
Multiple abscesses	48 (67.6)
Limited to the spleen on diagnosis	18 (25.3)
Lymph nodes	36 (50.7)
Skin	21 (29.5)
Liver	20 (28.1)
Lung	16 (22.5)
Muscle	7 (9.8)
Genitalia (vagina, prostate, testicles)	5 (7.0)
ENT	5 (7.0)
Kidney	5 (7.0)
Brain	4 (5.6)
Pancreas	4 (5.6)
Breast	2 (2.8)
Confirmation on histopathology, *n* (%)	59 (83.1)
Treatment, *n* (%)	
Antibiotics	59 (83.1)
Anti-tuberculosis drugs	11 (15.5)
Corticosteroids	61 (85.9)
Colchicine	27 (38.0)
Immunosuppressants/immunomodulators	44 (61.9)
Biologics	23 (32.3)
Relapse	44 (61.9)
Follow-up, med [IQR] in years (min-max)	6.8 (2.3;15.3) (1–30)

CRP: C-reactive protein; IQR: interquartile range; Med: median; PMN: polymorphonuclear neutrophils; SD: standard deviation.

**Table 2 jcm-11-03669-t002:** Diseases associated with the aseptic abscess syndrome and time to diagnosis.

	Total (*n* = 71)
Associated diseases, *n* (%)	41 (59.4)
Associated disease diagnosed concomitantly or after AA	23/41
Time to diagnosis of associated disease (years) vs. diagnosis of AA syndrome, med [IQR] (min; max)	0.4 (−2.0; +1.3) (−15.0; +25.0)
Subcategory of associated diseases, *n* (%)	
Inflammatory bowel disease	30 (42.2)
Crohn’s disease	26 (36.6)
Ulcerative colitis	4 (5.7)
Pyoderma gangrenosum	10 (14.3)
Relapsing polychondritis	6 (8.4)
Spondyloarthritis	3 (4.2)
Behcet’s disease	1 (1.4)
Rheumatoid arthritis	1 (1.4)
Time to diagnosis of associated disease (years), med [IQR]	
Time to diagnosis IBD vs. AA	0 (−0.5; +2.1)
Time to diagnosis PG vs. AA	1.0 (0; +1.6)
Time to diagnosis RP vs. AA	1.0 (−7.0; +1.1)
Time to diagnosis SPA vs. AA	1.6 (−5.2; +25.0)

AA: aseptic abscess; IBD: inflammatory bowel disease; IQR: interquartile range; Med: median; PG: pyoderma gangrenosum; RP: relapsing polychondritis; SPA: spondyloarthritis.

**Table 3 jcm-11-03669-t003:** Evolution among 44 patients suffering at least one aseptic abscess syndrome relapse.

	Total (*n* = 44)
Number of relapses, med (min-max) ±SD	1 (0–10)
1 relapse, *n* (%)	17 (38.6)
2 relapses, *n* (%)	11 (25.0)
> or =3 relapses, *n* (%)	16 (36.3)
Location of relapse vs. diagnosis, *n* (%)	
Same organ	32 (72.7)
Others organ	12 (27.3)
Relapse after splenectomy, *n* (%)	23 (100)
Location of relapse after splenectomy, *n* (%)	
Lymph nodes	7/23 (30.4)
Liver	6/23 (26.0)
Brain	4/23 (17.3)
Lung	4/23 (17.3)
Skin	3/23 (13.0)
Time to first relapse (years), med [IQR]	0.8 (0.5; 3.0)
Time to last relapse (years), med [IQR] (min; max)	3.5 (1.2; 9.0) (0.1–25.6)

IQR: interquartile range; Med: median; SD: standard deviation.

**Table 4 jcm-11-03669-t004:** Risk factors for aseptic abscess syndrome relapse in 71 patients.

	Univariate Analysis			Multivariate Analysis		
	HR	CI95%	*p*	HR	CI95%	*p*
Age ^1^	1.00	0.98–1.01	0.99			
Sex (female vs. male)	1.35	0.87–2.10	0.17			
Tobacco	1.61	0.84–3.08	0.15			
CRP ^1^	1.00	0.99–1.00	0.89			
Splenic abscess on diagnosis	0.90	0.55–1.46	0.68			
Hepatic abscess on diagnosis	1.59	1.05–2.42	0.02	2.14	1.35–3.40	0.001
Lymph node abscesses on diagnosis	1.36	0.88–2.10	0.15			
Skin abscesses on diagnosis	1.89	1.27–2.8	0.002	1.78	1.07–2.93	0.024
Aseptic abscess syndrome associated with another disease	0.68	0.41–1.11	0.125			
IBD	0.8	0.55–1.28	0.42	0.57	0.32–0.99	0.047
RP	0.8	0.6–1.17	0.31	0.61	0.29–1.28	0.19
Pyoderma gangrenosum	0.52	0.25–1.07	0.079	0.50	0.24–1.04	0.065
Colchicine	0.49	0.28–0.87	0.016	0.52	0.28–0.97	0.042
Biologics ^2^	1.91	1.11–3.28	0.018	1.2	0.61–2.36	0.59
Azathioprine	0.36	0.10–1.27	0.11			
Cyclophosphamide	1.39	1.07–1.8	0.12			
Splenectomy	0.8	0.5–1.35	0.71			

CI: confidence interval; IBD: inflammatory bowel disease; HR: hazard ratio; RP: relapsing polychondritis. ^1^ Variable treated as continuous data. ^2^ Biologics = infliximab, adalimumab, anakinra, ustekinumab, vedolizumab, and canakinumab.

## Data Availability

Not applicable.

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
