# Peer review of "Aseptic Abscess Syndrome: Clinical Characteristics, Associated Diseases, and up to 30 Years’ Evolution Data on a 71-Patient Series"

_jcm, 2022, doi:10.3390/jcm11133669_

Round 1

Reviewer 1 Report

Implement with a table that relates the type of immunomodulatory/immunosuppressant drugs (also combination of these types of drugs) practiced and the relapse of AA syndrome (also the sytes of the relapse). Better explained the results obtained, in the light of table to be inserted.

Reviewer 2 Report

Thank you for the opportunity to review this outstanding collation of information on a very important clinical entity that is often missed or is unfamiliar to clinicians (as evidenced by prolonged duration to diagnosis after initial presentation). 

1. I find the distinction between AA and the neutrophilic dermatoses (eg, pyoderma gangrenosum) to be confusing. When we reviewed the literature for our case (https://casereports.bmj.com/content/13/10/e236437), we did not find a lot of information in distinguishing the conditions. However, we did note that the histopathology seemed to be different between them: the deep abscesses of AAS are surrounded by a granulomatous reaction, which is absent in the classical form of PG. Did you look at this particular feature in your cases? It might be helpful to discuss a bit more the similarities and differences between AA and the neutrophilic dermatoses, especially if you are considering altering the diagnostic criteria to allow for isolated skin abscesses (in the absence of deep abscesses) as you allude to in your discussion.

2. In our case, the presence of pericardial and pleural friction rubs helped us establish the presence of a systemic inflammatory condition. Did you look at physical findings in these patients other than vital signs (eg, temperature, BP)? As a clinician, I would be curious to know if pericardial or pleural friction rubs are common in this condition.

3. In the case of AA, diagnosis is often delayed, usually as a result of unfamiliarity with the condition. In our case, the patient underwent weeks of hospitalization, including weeks of IV antibiotics, before she was transferred to our center and we were able to recognize the diagnosis (in large thanks to the previously excellent case series published by André, et al). There is tremendous cost to the patient when diagnosis is delayed, including both suffering as well as monetary loss (as well as to the healthcare system). It might be worthwhile to make some estimates about cost to the system in the cases you reviewed, including end points like days in the hospital, duration of unnecessary antibiotics, surgical procedures, etc.

Reviewer 3 Report

Reviewer comments

This is an interesting study and the authors have collected a unique dataset using cutting edge methodology. However, in my opinion the paper is still in need some modifications

Line

Comment

66

34.5 ±17 years [3–82]. No need for adding (min and max)

69

28,1%), and/or corticosteroids. add comma instead of and/or

70

and/or immunosuppressants…. add comma instead of and/or

77

colchicine appeared to be protective against relapse. this conclusion did not cope with the results mentioned

88

the disease. this disease

89

Only case reports have been published since the series of 2007 (3) and large data are lacking…need editing

105

Where is the ethical approval

151

The title of results is absent

153

Describe the data either in mean and SD or min and max

Too short description of the data ..only one line

Table 1

Number of organs involved during the course of the disease, mean ± SD (min-max)

It is more appropriate to be described in median ,min ,max

187

The paragraph is only one and half line which is not appropriate

Associated Conditions

This section needs editing

Pregnancy not comorbidity

The paragraphs are too short

221

16.2±8.47 ..it is not appropriate that the double  SD more than the mean .This is not accepted 

229

Rik…??

Number of relapses,

It is better to be described in med (min-max)

Other organ ..organs

Discussion ..need some editing to be more attractive to the authors

Round 2

Reviewer 1 Report

the changes made are adequate.

Reviewer 3 Report

I recommend acceptance of the paper in the present form